# Using Azadirachtin to Transform *Spodoptera frugiperda* from Pest to Natural Enemy

**DOI:** 10.3390/toxins13080541

**Published:** 2021-08-03

**Authors:** Sukun Lin, Shengnan Li, Zhenghui Liu, Li Zhang, Hao Wu, Dongmei Cheng, Zhixiang Zhang

**Affiliations:** 1Key Laboratory of Natural Pesticide and Chemical Biology of the Ministry of Education, South China Agricultural University, Guangzhou 510642, China; 13210615585@stu.scau.edu.cn (S.L.); wjyz@stu.scau.edu.cn (S.L.); zhangli0810@stu.scau.edu.cn (L.Z.); wuhao@stu.scau.edu.cn (H.W.); 2Guangdong Provincial Key Laboratory of Petrochemical Pollution Process and Control, Guangdong University of Petrochemical Technology, Maoming 525000, China; lzhhok@126.com; 3Department of Plant Protection, Zhongkai University of Agricultural and Engineering, Guangzhou 510225, China

**Keywords:** azadirachtin, maize, *Spodoptera frugiperda*, *Rhopalosiphum maidis*, natural enemy, pest control

## Abstract

*Spodoptera frugiperda* and *Rhopalosiphum maidis*, as main pests, seriously harm the safety of maize. At present, chemical pesticides are mainly used to control these pests. However, due to residue and resistance problems, more green, environmentally benign, simple preventive control technology is needed. In this study, we reported the reason for the antifeedant activity of azadirachtin on *S. frugiperda* and proposed that *S. frugiperda* treated with azadirachtin would turn from pest into natural enemy. *S. frugiperda* showed an obvious antifeeding phenomenon to maize leaf treated with various azadirachtin concentrations (0.5~20 mg/L). It was found that maize leaf treated with 1 mg/L of azadirachtin has a stimulating effect on the antenna and sensillum basiconicum of *S. frugiperda*, and azadirachtin can affect the feeding behavior of *S. frugiperda*. Additionally, after treating maize leaves or maize leaves + *R. maidis* with 1 mg/L of azadirachtin, the predatory behavior of *S. frugiperda* changed from a preference for eating maize leaves to *R. maidis*. Moreover, the molting of *R. maidis* can promote the change of this predatory behavior. Our results, for the first time, propose that the combined control technology of azadirachtin insecticide and biological control could turn *S. frugiperda* from pest into natural enemy, which can effectively eliminate *R. maidis* and protect maize. This combined control technology provides a new way for pest management and has good ecological, environmental, and economic benefits.

## 1. Introduction

Maize (*Zea mays* L.) is one of the three major food crops in the world, and it is also one of the most economically important crops in China [1]. Studies have suggested that the demand for maize will gradually increase with the rapid increase in China’s population and consequent economic growth [2]. Therefore, it is very important to ensure maize food security [3]. As the main pests on maize, *Spodoptera frugiperda* and maize aphid (*Rhopalosiphum maidis*) have seriously harmed the safety of maize [4,5].

*S. frugiperda* (J. E. Smith, 1797) is a kind of polyphagous migratory pest, which can cause considerable economic losses to more than 80 different crops [6]. This pest originated in tropical and subtropical areas of North, Central, and South America and has been regarded as the main pest of maize and several other crops for decades [7]. *S. frugiperda* was confirmed in several Asian countries including India, Sri Lanka, Thailand, Yemen, Myanmar, and Bangladesh in 2018 [8]. In China, this species was first found in the southeast of Yunnan Province in January 2019 and rapidly spread to most other provinces, causing serious damage to crops such as maize [9]. In 2018, published pest distribution and climatic suitability models indicated that there are environmental requirements for the permanent survival and reproduction of *S. frugiperda* in warmer regions of Asia [10]. Additionally, the geographical distribution model showed that most areas of China may be suitable for the survival of this destructive pest [11]. Farmers in the invaded areas are unprepared for this destructive pest, resulting in heavy losses and a sharp increase in the use of pesticides [12]. It is reported that farmers often use highly dangerous or prohibited pesticides and usually do not have appropriate personal protective equipment [13]. In addition to the strong ability of *S. frugiperda* to rapidly develop resistance to insecticides, overdependence on insecticides has become a major problem and will increase potential environmental and health risks [14]. Moreover, the large-scale application of pesticides increases the production cost. Farmers usually choose low-input methods to control pests and rely on natural biological control to a certain extent [15]. As a result, overdependence on insecticides is not in line with the integrated pest management (IPM) approach and is unsustainable in the long run. Currently, chemical control is still the main strategy against *S. frugiperda* in China, although some biological control experiments using predators have been conducted in the laboratory [16]. However, in the long run, more biological control strategies should be adopted against *S. frugiperda* to increase the sustainability of IPM programs.

Maize aphid (*R. maidis* Fitch), as another major pest on maize, is known to attack more than 30 genera and most cereal crops of Poaceae and was originally a species in Asia but is now distributed in tropical, subtropical, and warm temperate regions of the world [17]. *R. maidis* is harmful to the growth of maize by sucking the sap of the maize leaf [18]. In addition, it can spread the virus disease of the maize dwarf leaf, affect the photosynthesis of maize, cause the parasitism of mold, and reduce the yield in different degrees [19]. At present, the control of this pest mainly depends on strengthening field management and insecticides [20]. The use of insecticides aggravates the use and residues of pesticides and the cost of production, which undoubtedly needs to be improved to increase the sustainability of IPM programs.

Aiming to solve these problems more sustainably, we studied the effects of azadirachtin on the predation behavior of *S. frugiperda*. People have gradually shifted from the dependence on acute lethal effects to sublethal effects of insecticidal compounds, which is more in line with the sustainability of IPM programs [21]. Azadirachtin as a biopesticide, has a variety of resistance effects on many insects [22]. Additionally, previous research has shown that azadirachtin has repellent and antifeedant effects on many kinds of insects [23]. Biopesticides are usually unique and complex and have different modes of action, which means that pests are less likely to develop resistance, and biopesticides are often safer than chemical ones to the environment [24]. Azadirachtin arguably stands out as the most widely used biopesticide and is widely used in pest control [25].

In this study, we found that azadirachtin had antifeedant and repellent activities on *S. frugiperda*. Thus, we studied the effects of azadirachtin on the predation behavior of *S. frugiperda*. If azadirachtin treatment could change the predation behavior of the *S. frugiperda*, causing them to switch from eating maize leaf to preying on *R. maidis*, then it may be effective in killing *R. maidis*, significantly reducing the use of pesticides and protecting maize. Our study, for the first time, demonstrated that this combined control technology of azadirachtin insecticide and biological control could be considered as a new way for green, environmental protection and light control technology, with good ecological, environmental, and economic benefits.

## 2. Results

### 2.1. Toxicity of Azadirachtin to S. frugiperda

The results of the indoor toxicity of azadirachtin to *S**. frugiperda* are shown in Figure 1. It can be seen that after feeding maize leaves treated with different concentrations of azadirachtin (0.5~20 mg/L) for 3 days, the mortality rates varied significantly with respective concentrations. A mortality rate of 51.67% was evident when the larvae were fed with maize leaves treated with azadirachtin at 20 mg/L; the mortality rates gradually declined with the concentration decrease and became only 11.67% at the concentration of 1 mg/L. On the contrary, the mortality rate of the *S frugiperda* fed with maize leaves without azadirachtin only was 8.33%. In addition, it was found that the undead larvae of *S**. frugiperda* in the treatment groups appeared to be slow, repelled from treated leaves and refusing to feed on maize leaves containing azadirachtin. Additionally, it is difficult for azadirachtin to cause the death of *S. frugiperda* within two days.

### 2.2. Effects of Azadirachtin on the Antenna and Sensilla of S. frugiperda

A sublethal concentration of azadirachtin (1 mg/L) in maize leaves was selected for testing sublethal effects on *S**. frugiperda*. Based on the avoidance and antifeeding phenomena of the larvae in the toxicity test above, scanning electron microscopy was used to observe the antenna and sensilla of the fifth instar larvae fed with maize leaves treated with 1 mg/L of azadirachtin and without azadirachtin after 24 h. Compared with the larvae fed with maize leaves without azadirachtin, the antennae of the larvae fed with maize leaves treated with 1 mg/L azadirachtin retracted into the antennal nest, indicating that azadirachtin has a certain sensory effect on *S**. frugiperda* (Figure 2). In addition, further observation found that the sensillum basiconicum of the larvae fed with maize leaves treated with 1 mg/L azadirachtin showed obvious damage, such as fold and fracture (Figure 3). The results indicated that azadirachtin could stimulate the sensory organs of *S**. frugiperda*, which may affect the feeding behavior of *S**. frugiperda*.

### 2.3. Effects of Azadirachtin on Sensilla in the Feeding of S. frugiperda

The micro-drop method was used to further verify the effects of azadirachtin treatment on four sensilla in the feeding of *S**. frugiperda*. The cumulative feeding rates of the tested larvae on maize leaves changed into an “S” shape and increased with the increase of time within 48 h (Figure 4). At the same time point, the cumulative feeding rates decreased with the increase in azadirachtin concentration, and the cumulative feeding rates of the larvae fed with maize leaves containing azadirachtin were significantly lower than those of the larvae fed with maize leaves without azadirachtin, indicating that *S**. frugiperda* had an obvious antifeeding effect after sensillum basiconicum was treated with azadirachtin (Figure 4a). In Figure 4b, the cumulative feeding rates of the larvae feeding on maize leaves containing azadirachtin were significantly lower those that of the larvae feeding on maize leaves without azadirachtin after 16 h, indicating that azadirachtin treatment on sensillum styloconicum can also affect the feeding of *S. frugiperda*. In contrast, there was no significant difference in the cumulative feeding rates of the larvae feeding on maize leaves containing azadirachtin and the larvae feeding on maize leaves without azadirachtin at 8 and 16 h, indicating that there was no significant effect on the feeding of *S. frugiperda* when sensillum trichodeum or sensillum chaeticum was treated with azadirachtin (Figure 4c,d). These results showed that azadirachtin will affect the feeding of *S. frugiperda* after contact with the sensillum trichodeum or sensillum chaeticum.

### 2.4. Effects of Azadirachtin on Predatory Behavior of S. frugiperda

In the experiment of the effects of azadirachtin on the predatory behavior of *S. frugiperda* to *R. maidis*, the leaf consumption rates and number of aphids killed in 1 day of 3 groups are shown in Figure 5. On day 1, the leaf consumption rate of the ‘Leaf’ group was 5.00%, which was significantly lower than the ‘Leaf + Aphid’ group of 28.33%, and both groups were significantly lower than the control group of 66.67%, which showed that azadirachtin has an obvious antifeedant activity against *S. frugiperda*. On days 2 and 3, there was interestingly no significant difference between the leaf consumption rates of the ‘Leaf’ group and ‘Leaf + Aphid’ group, but both groups of leaf consumption rates were significantly lower than the control group (Figure 5a). After recording the number of aphids eaten in 1 day, we found that only 4.33 aphids were eaten in the control group on day 1, which was significantly lower than the ‘Leaf + Aphid’ group of 9.67, and the number of aphids eaten in ‘Leaf’ group was reached 17.67. On days 2 and 3, there was no significant difference between the number of aphids eaten in the ‘Leaf’ group and ‘Leaf + Aphid’ group, but both numbers of aphids eaten in the two groups were significantly higher than the control group (Figure 5b).

### 2.5. Effects of Aphid Molting on Predatory Behavior of S. frugiperda under Azadirachtin Treatment

The predation preference rates of *S. frugiperda* to *R. maidis* before and after aphid molting were tested and compared under 1 mg/L of azadirachtin treatment. Before aphid molting, the predation preference rates of *S. frugiperda* to aphid and maize leaf were 56.67% and 43.33%, respectively; however, the predation preference rates of *S. frugiperda* to aphid increased to 91.67% after aphid molting, and the predation preference rates of *S. frugiperda* to maize leaf decreased to only 9.33% (Figure 6). The results showed that aphid molting will increase the predation preference rate of *S**. frugiperda* to aphids and decrease the predation preference rate of *S**. frugiperda* to maize leaf.

## 3. Discussion

As an omnivorous invasive species, *S. frugiperda* causes serious damage to maize in China [26]. Traditional chemical control is prone to environmental pollution and resistance issues [27,28]. Maize producers need more greener, environmentally friendly, simple preventive control technology. Previous studies have shown that azadirachtin has a variety of biological effect on insects, such as anti-feeding, avoidance, cytotoxicity, and so on [29,30]. Due to its chemical complexity, azadirachtin has many anti-insect properties but no resistance problem [31]. In addition, this compound has also been reported as safer for non-target organisms [32]. In this study, the toxicity of azadirachtin against *S. frugiperda* was tested. From the experimental results of the toxicity test, little mortality of *S. frugiperda* was observed in the first two days after exposure to azadirachtin treated maize leaves. Additionally, the undead larvae of *S. frugiperda* appeared to exhibit obvious avoidance and antifeeding toward the maize leaves that contained azadirachtin. Therefore, a sublethal concentration of 1 mg/L was selected for subsequent experiments to highlight the sublethal effect of azadirachtin.

During the process of insect predation, the antenna and olfactory sensilla have an important role in the identification and positioning of the host plants, and insect sensilla can sensitively sense external signals, which is an important material basis for information exchange between insects and the outside world [33]. Thus, the study of antenna and sensilla can help us to understand the mechanism of the predation behavior of *S. frugiperda*. Different sensilla have different functions; it is generally believed that the sensillum trichodeum is mainly used to perceive and distinguish sex pheromones, and the sensillum chaeticum is mainly used to feel external mechanical stimuli [34]. In addition, the sensillum basiconicum is mainly used to feel the stimulation of plant volatiles, and the sensillum styloconicum mainly has the function of sensing taste stimulation [35]. Azadirachtin has antifeedant effect on *S. litura* and other lepidopterous larvae [36]. However, the target sensilla of azadirachtin on *S. frugiperda* larvae have not been reported. In view of this, the changes of sensilla and antennae of the fifth instar larvae of *S. frugiperda* fed with maize leaf treated with 1 mg/L of azadirachtin were observed after 24 h. Results showed that the antennae of *S. frugiperda* contracted into the antennal fossa (Figure 2), and the sensillum basiconicum was broken under azadirachtin treatment (Figure 3). This indicated that azadirachtin has a certain irritant effect on *S. frugiperda*, and the damage of sensillum basiconicum may affect the feeding behavior of *S. frugiperda*. In the experiment, four main sensilla were subsequently treated with azadirachtin, respectively, to determine the target sensilla and effect of azadirachtin on the feeding of *S. frugiperda* larvae. The results showed that sensillum basiconicum and sensillum styloconicum are the main target sensilla (Figure 4), which play an important role in the feeding process of *S. frugiperda*.

In the experiment of the effects of azadirachtin on the predatory behavior of *S. frugiperda* to *R. maidis*, it can be seen that after maize leaves were treated with 1 mg/L of azadirachtin, the *S. frugiperda* larvae showed obvious antifeeding phenomena, the leaf consumption rate on day 1 was only 5.00% compared to the control group of 66.67% (Figure 5a), and the larvae tended to prey on those foods that did not contain azadirachtin, such as aphids. When both maize leaves and aphids were treated with azadirachtin, the leaf consumption rate on day 1 was 28.33%, and 9.67 aphids were eaten by the larvae (Figure 5b); however, interestingly, the leaf consumption rate and the number of aphids eaten on day 2 changed significantly, the leaf consumption rate was decreased to 15%, and the number of aphids eaten by the larvae increased to 15.67. *R. maidis* larva has four instar stages and molts every 1~2 days [37,38]. Therefore, it is speculated that the molting of aphids may reduce azadirachtin residues and promote this change of predatory behavior. Taking maize leaves treated with azadirachtin as a comparison, the predation preference rates of *S. frugiperda* to *R. maidis* before and after aphid molting were measured and compared to determine the effect of aphid molting on the predatory behavior of *S. frugiperda* under azadirachtin treatment. After aphid molting, the predation preference rate of *S. frugiperda* to *R. maidis* was increased from 56.67% to 91.67% (Figure 6), which indicated that the molting of aphid can inhibit the effect of azadirachtin to the predatory behavior of *S. frugiperda*, which can achieve the purpose of protecting maize and preying on aphids, thereby reducing the application of pesticides.

In this study, azadirachtin could affect the feeding behavior of *S**. frugiperda*, causing it to prey on *R. maidis*; it may instead be effective in killing *R. maidis*, significantly reducing the use of pesticides and protecting maize. Additionally, the molting of aphids could promote the change in feeding behavior of *S**. frugiperda*. Our study, for the first time, demonstrated that this combined control technology of azadirachtin insecticide and biological control could be considered as a new way for green, environmental protection and light control technology, with good ecological, environmental, and economic benefits.

## 4. Conclusions

In this study, we determined the reason for the repellent and antifeedant activity of azadirachtin on *S. frugiperda* and proposed that *S. frugiperda* treated with azadirachtin would turn from pest into natural enemy. Azadirachtin has stimulating and damaging effects on the antennae and sensillum basiconicum of *S**. frugiperda* larvae and can affect the feeding behavior of *S. frugiperda*. When maize leaves or maize leaves + aphids are treated with azadirachtin, *S. frugiperda* will feed less on maize leaves and increase their predation preference rate for aphids, and the molting of aphids will promote a change of this feeding behavior. Our study combines azadirachtin insecticide with biological control and suggests that the combined control technology could be a new way for integrated pest management and has good ecological, environmental, and economic benefits. Future studies will focus on how to apply this combined control technology systematically to the field control of *S**. frugiperda*.

## 5. Materials and Methods

### 5.1. Plant, Insects, and Chemicals

The test maize (Corn Four) was planted in Zixi village, Huashan Town, Huadu District, Guangzhou, China (113°26′50.43″ N, 23°48′66.63″ E). No pesticides were sprayed before and during the experiment. In the fields, eggs of *S. frugiperda* were collected as the test source. After the eggs were hatched, they were fed with young maize leaves until 5th instar larvae for test. The feeding conditions were under controlled temperature (24~28 °C), relative humidity (55~65%), and photoperiod (16:8 L:D). *R. maidis* was also collected from the fields. Subsequently, the aphids were reared on maize leaves in a plastic culture dish (9 cm in diameter and 1 cm in height) in a growth chamber and maintained under controlled temperature (24~28 °C), relative humidity (55~65%), and photoperiod (12:12 L:D). To prevent malnutrition, fresh leaves were provided every 2~3 days.

Azadirachtin standard product (>90%) was kindly provided by Associate Professor Yongqing Tian of the Key Laboratory of Natural Pesticide and Chemical Biology of the Ministry of Education of South China Agricultural University. Azadirachtin was dissolved in acetone and the solution diluted to 100 mg/L in water and acetone (*v*/*v* = 30:70) and stored at 4 °C in freezer.

### 5.2. Toxicity of Azadirachtin to S. frugiperda

The objective of this experiment was to determine the sublethal concentration of azadirachtin to *S. frugiperda* for use in subsequent experiments. The azadirachtin standard solution was diluted in 70% (*v/v*) acetone aqueous solution to give concentrations of 20, 10, 5, 2, 1, and 0.5 mg/L, and 70% acetone aqueous solution was taken as control. All solutions were freshly prepared. In this experiment, adequate fresh maize leaves were soaked in different concentrations of azadirachtin solution for 3 s and dried naturally, and then put into a culture dish (9 cm in diameter and 1.5 cm in height), respectively, for moisturizing every day; finally, a 5th instar larva of *S. frugiperda* was put into each dish. Each treatment tested 20 larvae and was repeated 3 times. Three days after treatment, the survival state of the larvae in each treatment was examined. If the brush touched the surface of larva, no reaction was judged as death. The number of dead larvae was recorded, and the mortality of *S. frugiperda* was calculated based on the equation below:Mortality rate (%) = (m_1_/m_2_) × 100%
where m_1_ = the number of dead larvae, and m_2_ = the number of total larvae measured.

### 5.3. Effects of Azadirachtin on the Antennae and Sensilla of S. frugiperda

Based on the results of the above toxicity experiment, 70% acetone aqueous solution containing 1 mg/L azadirachtin was selected as sublethal level for the subsequent experiments. It was found that *S**. frugiperda* showed obvious avoidance and antifeeding phenomena to the maize leave which contained azadirachtins. In order to explore the reason, the changes of antennae and sensilla of the 5th instar larvae of *S. frugiperda* treated with azadirachtin were observed by scanning electron microscope. After the 5th instar larvae of *S**. frugiperda* were starved for 4 h, larvae in treatment group were reared with maize leaves treated with 1 mg/L azadirachtin, and larvae in control group were reared with maize leaves without azadirachtin. Each group tested 10 larvae and repeated 3 times. After 24 h of treatment, the larvae were immersed in phosphate-buffered solution, and their heads were cut off with a razor blade, placed in a centrifuge tube containing 5% glutaraldehyde, aspirated to precipitate, fixed for 24 h in 4 °C freezer, and taken out and washed with 0.1 mol/L phosphate buffer (pH = 7.2) 3 times, 20 min each time. Then, 30%, 50%, 70%, 80%, 90%, and 95% ethanol dehydration was performed, successively, 15 min each time, followed by dehydration with 100% ethanol 3 times, 20 min each time. The samples were dried at the critical point for 3 h, then stuck on the sample table with conductive glue, sprayed gold, and placed under scanning electron microscope. The accelerating voltage was 20 kV. Changes of antennae and sensilla were observed, and pictures were taken.

### 5.4. Treatment of Four Sensilla by Azadirachtin Solution

To further determine the cause of the influence of azadirachtin on the feeding behavior of *S**. frugiperda*, the micro-drop method was used to determine the feeding changes of the 5th instar larvae of *S. frugiperda* after the four sensilla (sensillum basiconicum, sensillum styloconicum, sensillum trichodeum, sensillum chaeticum) were treated with different concentrations (2, 1, and 0.5 mg/L) of azadirachtin. In this experiment, the larvae were starved for 4 h, and the four sensilla on the head surface of the larva were treated with azadirachtin solution, respectively. Each treatment was dropped with 0.1 μL, and each group was treated with 20 larvae, repeated 3 times. Afterwards, the larvae were placed in a culture dish with two maize leaves (5 cm × 5 cm) for feeding observation, respectively. Finally, the maize leaf area was counted at 8, 16, 24, 32, 40, and 48 h, and the cumulative feeding rate of the larvae was calculated based on the equation below:Cumulative feeding rate (%) = (n_1_/n_2_) × 100%
where n_1_ = the maize leaf area eaten by larva, and n_2_ = total maize leaf area.

### 5.5. Effects of Azadirachtin on Predatory Behavior of S. frugiperda

This experiment was divided into 3 groups to determine the effects of azadirachtin on predatory behavior of *S. frugiperda*. In first group, which is called ‘Leaf’, fresh maize leaves were cut into 5 cm × 5 cm, soaked in 1 mg/L azadirachtin solution for 3 s, and dried naturally. Then, a leaf was placed in a culture dish (9 cm in diameter and 1.5 cm in height), 20 young aphids with 70% acetone aqueous solution sprayed and dried were put into the dish, and finally, a 5^th^ instar larva was introduced. Leaf and aphids were replaced every day. In second group, which was called ‘Leaf + Aphid’, aphids were sprayed with 1 mg/L azadirachtin solution, and the other steps were the same as those of first group. In control group, the leaves were soaked in 70% acetone aqueous solution, the aphids were sprayed with 70% acetone aqueous solution, and the other steps were the same as those of first group. Each group tested had 20 larvae, and tests were repeated 3 times. The number of aphids eaten every day was recorded. Additionally, the leaf consumption rate was calculated using the equation below:Leaf consumption rate (%) = (p_1_/p_2_) × 100%
where p_1_ = the maize leaf area eaten by larva, and p_2_ = maize leaf area.

### 5.6. Effects of Aphid Molting on Predatory Behavior of S. frugiperda under Azadirachtin Treatment

The predation selectivity of *S. frugiperda* to *R. maidis* was tested and compared before and after aphid molting to determine the effects of aphid molting on predatory behavior of *S. frugiperda* under azadirachtin treatment; maize leaves treated with 1 mg/L azadirachtin solution were used as a comparison. In this experiment, maize leaf was soaked in 1 mg/L azadirachtin solution and dried naturally, then placed in a culture dish (9 cm in diameter and 1.5 cm in height). At the same time, 20 aphids sprayed with 1 mg/L azadirachtin solution then dried were introduced, and finally, a 5^th^ instar larva was introduced after starved for 4 h. Then, we observed whether the larva consumed the leaf or aphid first and recorded the number of larvae which selected an aphid first. In order to test the predation preference rates of *S. frugiperda* to *R. maidis* after aphid molting, 20 aphids sprayed with 1 mg/L azadirachtin solution and dried were put into a culture dish with azadirachtin-treated leaves after molting. Then, a 5^th^ instar larva was introduced after starved for 4 h. Finally, we observed whether the larva consumed the leaf or aphid first and recorded the number of larvae which first selected to eat an aphid. Twenty larvae were tested, and tests were repeated 3 times. The predation preference rates to *R. maidis* were calculated using the equation below, and the predation preference rates before and after aphid molting were compared:Predation preference rate (%) = (q_1_/q_2_) × 100%
where q_1_ = the number of larvae first selecting to eat aphid, and q_2_ = the number of total larvae measured.

### 5.7. Data Analysis

Collected data from the aforementioned experiments were subjected to one-way analysis of variance (ANOVA) using SPSS Statistics, Version 17.0, 2009 (International Business Machines Corporation, Armonk, NY, USA). If significant differences occurred among treatments, means were separated by Tukey’s honestly significant difference (HSD) test at *p* < 0.05 level. Means were presented in graphs with standard error, which were drawn using Microsoft Excel.

## Figures and Tables

**Figure 1 toxins-13-00541-f001:**
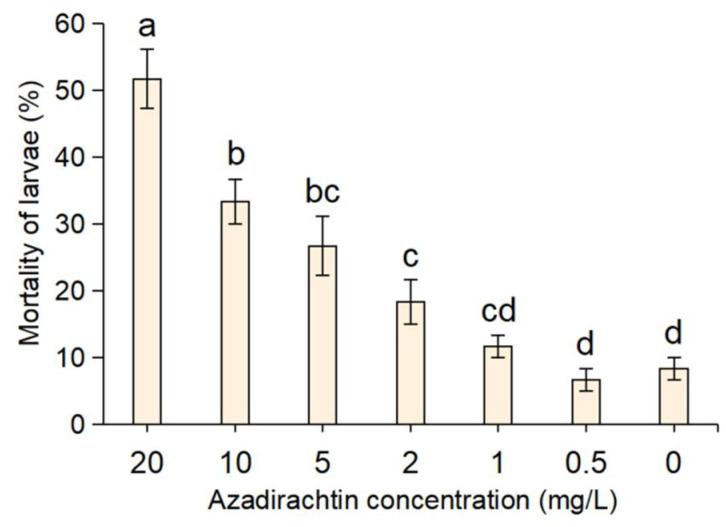
Mortality rates of *S**. frugiperda* after three days of feeding with maize leaves treating different concentrations of azadirachtin. The ‘0′ on the *x*-axis was the control, which means maize leaves without azadirachtin. Data are presented as mean ± standard error (S.E.). Different letters above bars indicate significant differences in mortality among treatments due to concentration effects at *p* < 0.05 level based on Tukey’s honestly significant difference (HSD) test (*n* = 3).

**Figure 2 toxins-13-00541-f002:**
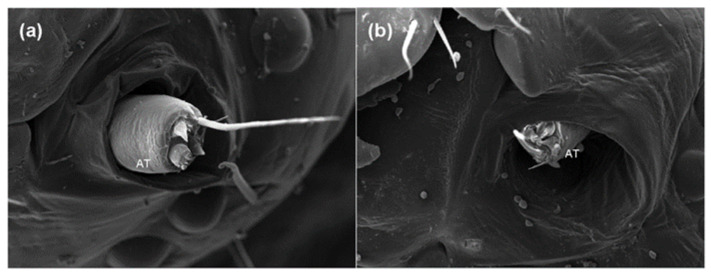
Morphological differences in the antenna of the 5th larvae of *S**. frugiperda* after feeding on maize leaves without azadirachtin (**a**) and treated with 1 mg/L of azadirachtin (**b**); AT: Antenna.

**Figure 3 toxins-13-00541-f003:**
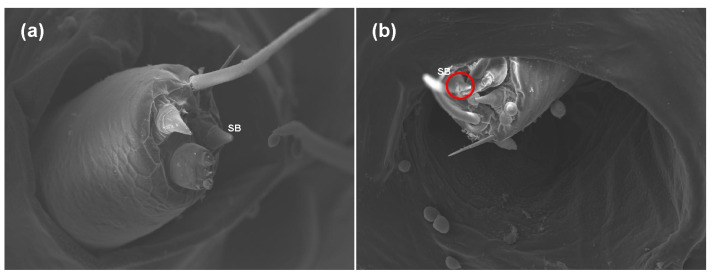
Morphological differences in the sensilla of the 5th larvae of *S. frugiperda* after feeding on maize leaves without azadirachtin (**a**) and treated with 1 mg/L of azadirachtin (**b**); SB: sensillum basiconicum.

**Figure 4 toxins-13-00541-f004:**
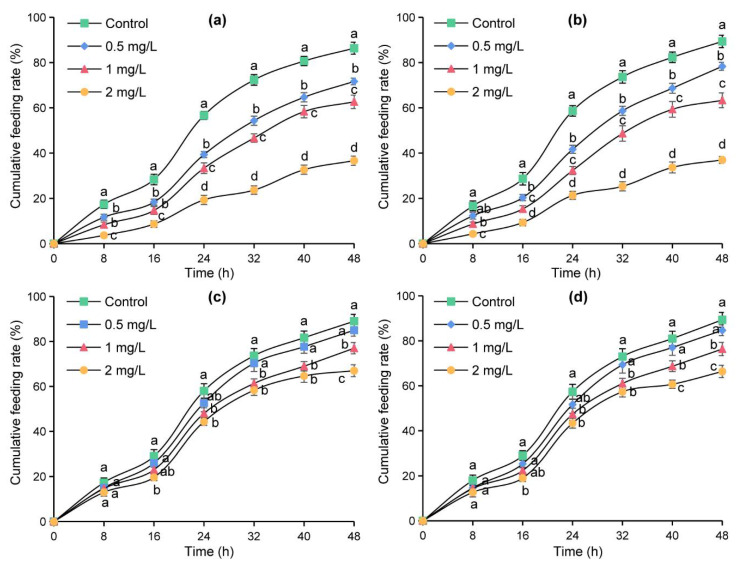
Cumulative feeding rates of *S**. frugiperda* after treatment with different concentrations of azadirachtin on sensillum basiconicum (**a**), sensillum styloconicum (**b**), sensillum trichodeum (**c**), and sensillum chaeticum (**d**), respectively. Data are presented as mean ± standard error (S.E.). Different letters at each recording time point indicate significant differences among treatments at *p* < 0.05 level based on Tukey’s HSD test (*n* = 3).

**Figure 5 toxins-13-00541-f005:**
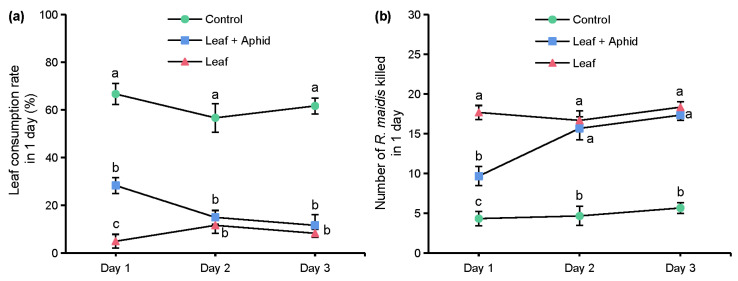
Leaf consumption rate in 1 day (**a**) and number of *R. maidis* killed in 1 day (**b**). ‘Leaf’ indicates leaves soaked with 1 mg/L of azadirachtin solution and aphids sprayed with 70% acetone aqueous solution. ‘Leaf + Aphid’ indicates treating leaves and aphids with 1 mg/L of azadirachtin solution. ‘Control’ indicates leaves and aphids treated with 70% acetone aqueous solution. Data are presented as mean ± S.E. Different letters at each sampling day indicate significant differences among treatments at *p* < 0.05 level based on Tukey’s HSD test (*n* = 3).

**Figure 6 toxins-13-00541-f006:**
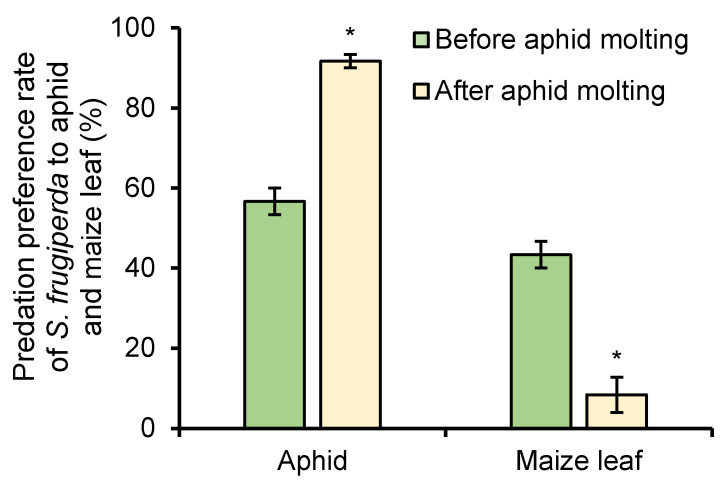
Predation preference rate of *S. frugiperda* to aphid and maize leaf before and after aphid molting under 1 mg/L of azadirachtin treatment. Data are presented as mean ± S.E. ‘*’ represents significant differences between two situations at *p* < 0.05 level based on Tukey’s HSD test (*n* = 3).

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
