# Peer review of "Using Azadirachtin to Transform Spodoptera frugiperda from Pest to Natural Enemy"

_toxins, 2021, doi:10.3390/toxins13080541_

Round 1
Reviewer 1 Report
The English language of the manuscript must be improved: many sentences are difficult or even impossible to understand.
The azadirachtin treatment method (spraying with solutions containing 70% acetone) raises the possibility of damage of the treated organism (either the plant, the aphid or the armyworm larvae) caused by the solvent itself. A reference of this method proving that it is harmless must be included (line 258). It would be even better to include a control with acatone solvent and a control without it.
Some references are missing in the text, replaced with 'xxx' characters (lines 251, 334, 335).
The valid name of the plant family containing the most important cereals is Poaceae (Gramineae is outdated, line 53).
In Fig. 1., the azadirachtin concentration should be in increasing order (from left to right). The quality of the graph must be improved. Also, the quality (resolution) of the Fig. 4., 5., 6. must be improved, some letters indicating the significant difference can be hardly read.
Fig. 2. and Fig. 3. seem identical, the latter with a narrower cropping and lower resolution. The two should be replaced with one.
Author Response
Response to Reviewer 1 Comments
Point 1: The English language of the manuscript must be improved: many sentences are difficult or even impossible to understand.
Response 1: Thanks for your value comments. We have carefully addressed your comments. The modifications are marked in red in this manuscript.
Point 2: The azadirachtin treatment method (spraying with solutions containing 70% acetone) raises the possibility of damage of the treated organism (either the plant, the aphid or the armyworm larvae) caused by the solvent itself. A reference of this method proving that it is harmless must be included (line 258). It would be even better to include a control with acatone solvent and a control without it.
Response 2: According to the results of preliminary experiments, 70% acetone aqueous solution has no significant damage effect on the treated organism (either the plant, the aphid or the armyworm larvae). And the reference of this method has been added in the manuscript.
Point 3: Some references are missing in the text, replaced with 'xxx' characters (lines 251, 334, 335).
Response 3: Due to journal requirements, the information related to the authors has been temporarily hidden.
Point 4: The valid name of the plant family containing the most important cereals is Poaceae (Gramineae is outdated, line 53).
Response 4: It has been modified in the manuscript.
Point 5: In Fig. 1., the azadirachtin concentration should be in increasing order (from left to right). The quality of the graph must be improved. Also, the quality (resolution) of the Fig. 4., 5., 6. must be improved, some letters indicating the significant difference can be hardly read. Fig. 2. and Fig. 3. seem identical, the latter with a narrower cropping and lower resolution. The two should be replaced with one.
Response 5: The figures have been modified in the manuscript.
Reviewer 2 Report
The manuscript presented original work that will be of interest to those working in pest management. In terms of general comments, I am concerned that the authors did not address whether they tested for or had problems with homogeneity of variance when applying ANOVA to the percentile data. Variances will vary tremendously over the range of values. The other comment may be a language issue in that they state repeatedly leaves containing azadirachtin at different concentrations. But this was never measured. What they did was to soak leaves in different azadirachtin concentrations which is very different. They need to change these reference to 'leaves treated with X mg/L azadirachtin. In general the readability of the text should be improved. Here are my specific comments:
P1, L4: Delete 'have'.
P1, L6: Change to read 'due to the residue and resistance problems, more green, environmentally benign, simple preventive and control technology is needed.'
P1, L33: Delete 'intrusion'.
P1, L34: Change 'such as' to 'including'.
P2, L46-47: Sentence is incomplete.
P2, L51 & 59: change 'under the perspective of IPM.' to 'to increase the sustainability of IPM programs.'
P2, L62-3: Reword sentence, incomplete.
P2, L66: Changes 'ways' to 'mode.'
P2, L67: Change 'safer' to 'often safer'.
P2, L86: Change 'repellent' to 'repelled from treated leaves'.P3: L102: Change 'stimulating' to 'sensory'.
P3, Figure 1: X-axis title is azadirachtin concentration, I feel this should this should include 'of treatment solution' as this can be confused with leaf concentration.
P3, L105: Change 'simulated' to 'sensory'.
P4, Figure 3: SB not referenced in the photos. Need to add this or drop from figure caption. Fig 3 is the same as fig 2, just enlarged with the red circle, can this be combined with figure 2?
P5, Figure 4c: Hard to read HSD letters at 8 and 16 hours.
P6, Figure 5b: The y-axis should stop at 20 as this would be the total number of aphids in a dish. Much better explanation of azadirachtin concentrations in this caption.
P6, L160: The authors refer to 'predation rates' but these are not predation rates. They are predation preference rates. A predation rates would be the number of aphids eaten per day for example, not the number of larvae chosing to eat an aphid first.
P6, Figure 6: Add 'aphid' to legend before molting.
P6, L174: change to read' Maize producers need more greener, enviromentally friendly, simple preventive and control technology.'
P7, L176: Change end of sentence to read 'avoidance and cytotoxicity.'
P7, L181: Change sentence to read 'Little mortality of S. frugiperda was observed in the first two days after exposure to azadirachtin treated maize leaves.
P7, L185: Good paragraph!
P7, L214: Change to read '... of aphid may reduce azadirachtin residues and promote this change of predatory behavior.'
P7, L217: Azadirachtin misspelled.
P8, L245: Insert 'they were' before 'fed with'.
P8, L250: Change to read '... prevent malnutrition, fresh laves were provided every 2~3 days.'
P9, L299: Chnage to read '... which is called Leaf.'
P10, L 317: Change to read ' ... whether the larva consumed the leaf or aphid first, and recorded the number of larvae which selected an aphid first.'
P10, L322: Change to read '... the larva consumed the leaf or aphid first, and recorded ...'
P10, (4): Change to read 'Selective predation preference rate (5)'.
P10, L328: How was homogeneity of variance evaluated when analyzing the percentage data?
Author Response
Response to Reviewer 2 Comments
Point 1: The manuscript presented original work that will be of interest to those working in pest management. In terms of general comments, I am concerned that the authors did not address whether they tested for or had problems with homogeneity of variance when applying ANOVA to the percentile data. Variances will vary tremendously over the range of values. P10, L328: How was homogeneity of variance evaluated when analyzing the percentage data?
Response 1: Thanks for your constructive comments. The suggestions are addressed in the following. We converted the percentage data into numbers and then performed analysis of variance.
Point 2: The other comment may be a language issue in that they state repeatedly leaves containing azadirachtin at different concentrations. But this was never measured. What they did was to soak leaves in different azadirachtin concentrations which is very different. They need to change these references to 'leaves treated with X mg/L azadirachtin.
Response 2: They have been modified in the manuscript, and the modifications are marked in red.
Point 3: P4, Figure 3: SB not referenced in the photos. Need to add this or drop from figure caption. Fig 3 is the same as fig 2, just enlarged with the red circle, can this be combined with figure 2?
Response 3: Figure 2 demonstrates the changes in the antennae of Spodoptera frugiperda before and after azadirachtin treatment, and Figure 3 is to demonstrate the changes in the sensilla of Spodoptera frugiperda before and after azadirachtin treatment. Thus, if the two figures are combined, it is not convenient to demonstrate the changes in the antennae and sensilla of Spodoptera frugiperda before and after azadirachtin treatment.
Point 4: In general the readability of the text should be improved. Here are my specific comments:
P1, L4: Delete 'have'.
P1, L6: Change to read 'due to the residue and resistance problems, more green, environmentally benign, simple preventive and control technology is needed.'
P1, L33: Delete 'intrusion'.
P1, L34: Change 'such as' to 'including'.
P2, L46-47: Sentence is incomplete.
P2, L51 & 59: change 'under the perspective of IPM.' to 'to increase the sustainability of IPM programs.'
P2, L62-3: Reword sentence, incomplete.
P2, L66: Changes 'ways' to 'mode.'
P2, L67: Change 'safer' to 'often safer'.
P2, L86: Change 'repellent' to 'repelled from treated leaves'.
P3: L102: Change 'stimulating' to 'sensory'.
P3, Figure 1: X-axis title is azadirachtin concentration, I feel this should this should include 'of treatment solution' as this can be confused with leaf concentration.
P3, L105: Change 'simulated' to 'sensory'.
P5, Figure 4c: Hard to read HSD letters at 8 and 16 hours.
P6, Figure 5b: The y-axis should stop at 20 as this would be the total number of aphids in a dish. Much better explanation of azadirachtin concentrations in this caption.
P6, L160: The authors refer to 'predation rates' but these are not predation rates. They are predation preference rates. A predation rates would be the number of aphids eaten per day for example, not the number of larvae chosing to eat an aphid first.
P6, Figure 6: Add 'aphid' to legend before molting.
P6, L174: change to read' Maize producers need more greener, enviromentally friendly, simple preventive and control technology.'
P7, L176: Change end of sentence to read 'avoidance and cytotoxicity.'
P7, L181: Change sentence to read 'Little mortality of S. frugiperda was observed in the first two days after exposure to azadirachtin treated maize leaves.
P7, L185: Good paragraph!
P7, L214: Change to read '... of aphid may reduce azadirachtin residues and promote this change of predatory behavior.'
P7, L217: Azadirachtin misspelled.
P8, L245: Insert 'they were' before 'fed with'.
P8, L250: Change to read '... prevent malnutrition, fresh laves were provided every 2~3 days.'
P9, L299: Change to read '... which is called Leaf.'
P10, L 317: Change to read ' ... whether the larva consumed the leaf or aphid first, and recorded the number of larvae which selected an aphid first.'
P10, L322: Change to read '... the larva consumed the leaf or aphid first, and recorded ...'
P10, (4): Change to read 'Selective predation preference rate (5)'.
Response 4: They have been modified in the manuscript, and the modifications are marked in red. Thank you again for your careful and constructive comments.
Reviewer 3 Report
First of all, structure must be changed. The Material and Methods section must be placed after Introduction and before the Results section.
I think the main point (not enough highlighted in the title, discussion and conclusions) is the finding that S. frugiperda could behave as a predator. But even it could be the first time this fact is reported for S. frugiperda, at least the authors should include references on similar cases in the Discussion.
Please pay attention to English language and style. Some sentences are nonsense or gramatically incorrect.
Lines 46-47: "Farmers....", the comma must be deleted or something changed to have sense
Line 51: delete "to" in "adopted to against"
Lines 52-54: Sentence too long
Line 55: write "sap" instead of "juice"
Line 58: at the end, "residues" (the s is missing)
Lines 69-...: In this section the authors must not talk about the results, but the objectives of the paper.
Lines 87-88: "and it is difficult...". This phrase is not supported by any reference.
Lines 97-100: This phrase must be in the Material and Methods section.
Line 105: "could stimulate" (without final d), "which may affect..." (this is speculation)
Lines 115-116: This phrase must be in the Material and Methods section.
Discussion section: This section is not intended to present again the results but to compare the present findings with those obtained by other authors and DISCUSS why are similar or different. Any bibliographic reference has been included. Please search for similar works, compare, discuss, give support to your findings...
Line 241: Plants, insects and chemicals (not Plants, Insects and Chemicals)
Line 245: "...were hatched, THEY WERE fed with young maize leaves..."
Line 246: NEVER use abbreviation at the beginning of a sentence
Line 250: Write "fresh leaves were replaced" instead of "replaced with fresh leaves"
Line 251: xxx and xxx ?????
Lines 251-253: revise English.
Lines 257-258: revise English.
Line 261: "for moisturizing every day"?????
Lines 276-282: Sentence too long.
Lines 283-284: revise English.
Section 5.5: Why this experiment only was carried out for 3 days?
Lines 307-308: revise English.
Section 5.6: If molting is so often (2 days) maybe this effect is not very significant...
Lines 311-312: Delete from "The predation selectivity..." to "aphid molting"
Lines 334- 334: xxxx ??????
Author Response
Response to Reviewer 3 Comments
Point 1: First of all, structure must be changed. The Material and Methods section must be placed after Introduction and before the Results section.
I think the main point (not enough highlighted in the title, discussion and conclusions) is the finding that S. frugiperda could behave as a predator. But even it could be the first time this fact is reported for S. frugiperda, at least the authors should include references on similar cases in the Discussion.
Please pay attention to English language and style. Some sentences are nonsense or gramatically incorrect.
Response 1: Thanks for your comments. We have optimized the manuscript and the modifications were marked in red.
Point 2: Line 251: xxx and xxx ????? Lines 334- 334: xxxx ??????
Response 2: Due to journal requirements, the information related to the authors has been temporarily hidden.
Point 3:
Lines 46-47: "Farmers....", the comma must be deleted or something changed to have sense
Line 51: delete "to" in "adopted to against"
Lines 52-54: Sentence too long
Line 55: write "sap" instead of "juice"
Line 58: at the end, "residues" (the s is missing)
Lines 69-...: In this section the authors must not talk about the results, but the objectives of the paper.
Lines 87-88: "and it is difficult...". This phrase is not supported by any reference.
Lines 97-100: This phrase must be in the Material and Methods section.
Line 105: "could stimulate" (without final d), "which may affect..." (this is speculation)
Lines 115-116: This phrase must be in the Material and Methods section.
Discussion section: This section is not intended to present again the results but to compare the present findings with those obtained by other authors and DISCUSS why are similar or different. Any bibliographic reference has been included. Please search for similar works, compare, discuss, give support to your findings...
Line 241: Plants, insects and chemicals (not Plants, Insects and Chemicals)
Line 245: "...were hatched, THEY WERE fed with young maize leaves..."
Line 246: NEVER use abbreviation at the beginning of a sentence
Line 250: Write "fresh leaves were replaced" instead of "replaced with fresh leaves"
Lines 251-253: revise English.
Lines 257-258: revise English.
Lines 276-282: Sentence too long.
Lines 283-284: revise English.
Lines 307-308: revise English.
Section 5.6: If molting is so often (2 days) maybe this effect is not very significant...
Lines 311-312: Delete from "The predation selectivity..." to "aphid molting"
Response 3: They have been modified in the manuscript, and the modifications are marked in red. Thank you again for your careful comments.
Point 4: Line 261: "for moisturizing every day"?????
Response 4: Maize leaves will be dry if they are not moisturizing. Spodoptera frugiperda and Rhopalosiphum maidis will not feed on dry leaves.
